# Homeopathy and Psychological Therapies

**Davide Donelli** * and **Michele Antonelli**

AUSL-IRCCS Reggio Emilia, 42122 Reggio Emilia, Italy; michele.antonelli@ausl.re.it
* Correspondence: davide.donelli@ausl.re.it

**Definition:** Homeopathy is a popular, although highly debated, medicinal practice based on the administration of remedies in which active substances are so diluted that no detectable trace of them remains in the final product. This hypothesis paper aims to outline a possible reinterpretation of homeopathy in the light of psychological therapies in order to improve its clinical safety and sustainability.

**Keywords:** homeopathy; psychology; reinterpretation; hypothesis



## 1. Introduction

Homeopathy is a popular, although highly debated, medicinal practice. In Italy, for example, it is estimated that, even if with a slightly declining trend, around 4.1% of the entire population (almost 2.5 million people) occasionally or regularly seeks homeopathic care, and these data, collected in 2013, suggest that homeopathy is the most used Complementary and Alternative Medicine (CAM) by Italians [1]. Epidemiological studies aimed to assess the worldwide prevalence of homeopathy use have reported similar data for other high-income countries [2].

Homeopathy was first invented by the German doctor Samuel Hahnemann (1755–1843), and it is based on the administration of remedies in which active substances are so diluted that no detectable trace of them remains in the final product [3]. In his empirical studies, Hahnemann reported that the self-administration of a common antimalarial medicinal plant (*Cinchona*) resulted in the occurrence of the same symptoms of malaria, but to a milder degree [4]. This led him to cast the foundations of a new medicinal system called "homeo-pathy", a noun that originates from the combination of two Greek terms: "homeo-" (from "homoios", a prefix standing for "same", "like") and "-pathy" (from "patheia", or "suffering", a suffix usually indicating all diseases) [5]. Hahnemann stated that the proper disease remedy has to be chosen by the homeopathic practitioner on the basis of the principle summarized by a Latin expression: "similia similibus curantur" (literally, like cures like) [3]. In other words, a therapeutic remedy is recommended for a given illness if pharmacological doses of the original substance would theoretically produce the same effects on the body as the disease symptoms. The preparation of a homeopathic remedy implies a serial high dilution of the original substance, along with some mechanical "succussions" or shakes (a process sometimes called "dynamization" or "potentiation", which is believed to boost the efficacy of the remedy). Every homeopathic product is usually labelled with a Latin name (for example: *Aconitum napellus*) indicating the original principle which has undergone serial dilutions. Latin is used because of historical traditions and because all physicians worldwide can more easily and unequivocally understand it [6]. Along with their name, homeopathic products often carry a brief description of "how much" they have been diluted: For example, "30 CH" means that the original principle has been diluted by a factor of 100 (usually in a hydroalcoholic solution) for 30 times. More precisely, "CH" stands for "Hahnemann's Centesimal [dilution]", thus referring to the method for preparation of homeopathic remedies invented by the German doctor [6]. Common formulations of homeopathic products found in the market usually include sucrose-

and/or lactose-containing granules, globules (smaller than granules), and liquid drops for oral consumption; other formulations for local applications like creams, ointments, eye-drops, and nasal sprays are also available (Figure 1) [7].

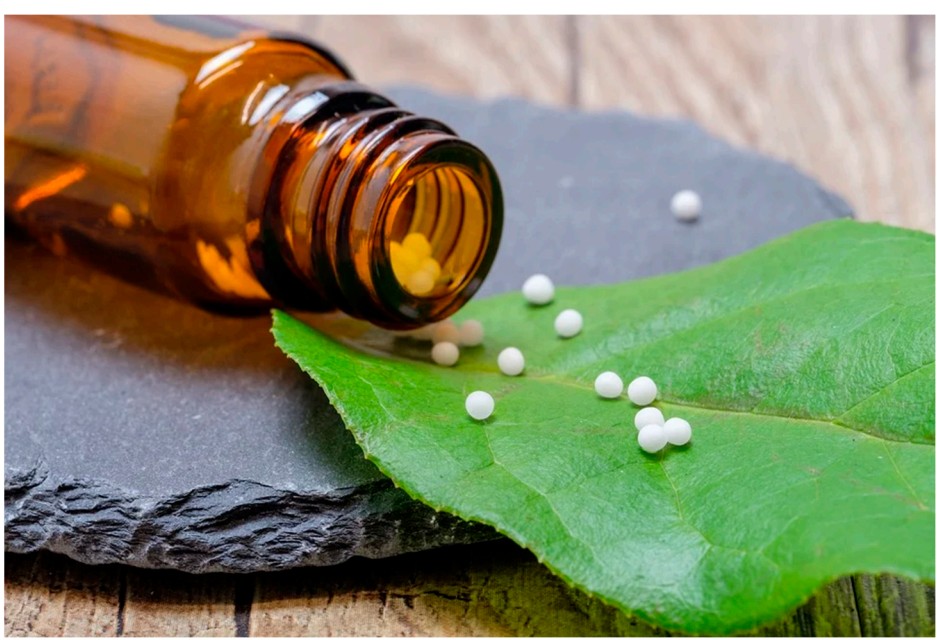

**Figure 1.** A homeopathic remedy. Homeopathic remedies are often produced in the form of granules or globules to be taken orally. From: https://pixabay.com/cs/photos/globuli-homeopatie-naturopatii-3163134/.

Originally, homeopathy was proposed for the treatment of any disease, and quickly gained popularity, with one of the first institutions dedicated to it being founded in the USA as early as in the last part of the nineteenth century (the American Institute of Homeopathy) [4]. Since then, various homeopathic clinics and hospitals, even with inpatient services, have been created all around the world: However, in Western countries like the UK, the public financial support of these facilities has mostly been withdrawn in recent years and the acceptability of this alternative medicinal system in public-funded healthcare systems has been questioned openly [4,8]. Similarly, in France, it was decided that the cost of homeopathic remedies is not to be covered by the national fund for health as of 2021 [9]. All the same, there are European countries, such as Luxembourg, where homeopathic remedies are still reimbursed by public health insurances [9]. In Italy, although homeopathic remedies are to be paid for by individual patients in an "out-of-pocket" fashion, there are a few outpatient clinics where the cost of homeopathic consultations is partially or fully subsidized by the public healthcare system, depending on personal income [10]. The case of Switzerland, in which homeopathy, after being withdrawn from the list of reimbursed medicines in 2005, was reintroduced as a basic health insurance-subsidized practice with a general referendum in 2009, shows that, beyond the scientific debate, popular support plays an important role in regulating homeopathic practice [11]. In general, worldwide, specific regulations of homeopathy largely vary on the basis of national laws, as described in a detailed report issued by the World Health Organization in 2001 [12].

The debate around homeopathy has a long history and almost dates back to its origin. After years of speculations, discussion, research, and even fights and controversies, in 2005, a famous editorial published in "The Lancet" concluded that, in consideration of available scientific evidence, the history of homeopathy had eventually come to an end [13], and briefly mentioned the main results of a large study conducted by Aijing Shang et al., where the authors described their findings as "compatible with the notion that the clinical effects of homoeopathy are placebo effects" [14]. However, some supporters of homeopathy

raised objections against Shang's study methods and conclusions [15], and other researchers further prompted the original debate by underscoring that misunderstandings in the definition of homeopathy (as a therapeutic system, as a highly-diluted remedy, as the treatment by a homeopath, or as the inspiring principles of this practice) have resulted in a potentially biased assessment of its therapeutic effects within trial settings [16]. In particular, it was suggested that studying the pharmacological action of homeopathic remedies as if they were drugs does not account for the benefits of the patient–homepath relationship and, therefore, trials aimed to evaluate the overall clinical efficacy of this practice should be designed in such a way as to consider even this less standardizable and harder-to-measure aspect [16]. However, other researchers underscored that homeopathy is to be dismissed, due to its unproven efficacy, dangerous uses as an alternative medicinal system, and basic foundations of obsolete metaphysical concepts [17,18]. In general, the debate has continued over the years, and homeopathy has alternatively been portrayed both as quackery to be banned and as a valid therapeutic practice with a future within standard medicine [19]. Still today, homeopathy remains popular, and the debate around its efficacy and usefulness appears far from being closed [20].

On the one hand, as illustrated by its supporters, some reasons for preserving homeopathy are the following:

- Homeopathy, especially when "individualized" (namely tailored to the patient's and disease characteristics), has some effects beyond placebo, and this is also confirmed by the experience of several patients and practitioners [21,22].
- Homeopathy is not only useful in medicine, but even in other fields of biological science, as it may also have a role for plant health [23].
- Homeopathic remedies are safe and, in general, highly tolerated by patients [24].
- There are some accredited universities and colleges where homeopathy is taught [25].

On the other hand, some reasons against homeopathy are the following:

- The efficacy of homeopathy is scientifically unproven (both in human and in veterinary medicine) and demonstrated only for self-limiting or placebo-responsive health conditions [14,26,27].
- The basic principles of this practice are not compatible with modern chemistry and physics, and appear to be more philosophical than science-based [18].
- Homeopathy can be dangerous when embraced as an alternative to standard medicine, especially for severe and life-threatening diseases [28–30].
- There are no recent advances in homeopathy, as its body of knowledge seems not to make any relevant progress over the decades [17].

To date, available high-quality scientific evidence from existing umbrella reviews shows that homeopathy efficacy isn't clearly superior to placebo, even when considering individualized homeopathy, in which the remedy is chosen not only on the basis of the disease type, but also depending on all subjective symptoms/manifestations reported by the patient [26,31,32]. Extensive literature reviews from the Australian Health and Medical Research Council, from the British National Healthcare System, and from the European Academies' Scientific Advisory Council point towards the same direction [33–35]. This compelling evidence basis is justifiably pushing both the scientific community and policymakers to refuse and dismiss homeopathy as a medicinal practice, thus ultimately hoping for its official ban, with information and educational campaigns pointing towards this objective. However, there is still a part of the population that is fond of homeopathy because of its subjectively perceived beneficial effects (actually, it is not possible to raise objections on individually reported benefits, even if caused by a placebo), and there are also homeopathic physicians and healthcare workers who feel under professional attack when attempts to ban homeopathy are made. These conflicts fuel tensions, separations, and radicalization of opposed beliefs, with no or poor chances to establish a fruitful dialogue [36,37].

An obstacle to an open dialogue is probably the earliest interpretation of homeopathy, which was originally conceived as an alternative medicinal system capable of treating any disease or health-related complaint. This interpretation can be dangerous and has to be rejected: In fact, several cases of patients affected by severe health conditions and experiencing poor clinical outcomes (or even death) after refusing standard care and opting for homeopathy have been reported both by newspapers and in the scientific literature [28–30]. In order to create a bridge between homeopathy and standard medicine, it is necessary to reinterpret homeopathy in such a way as to make it become a sustainable and safe practice, in accord with available scientific evidence.

## 2. Scientific Evidence and Placebo Effects

In a recent systematic literature review summarizing evidence from over 60 reviews, the efficacy of homeopathy was analyzed along with effects of open-label placebo treatments (namely the administration of a pharmacologically inert substance without deception) in order to find a possible way to reinterpret homeopathy in the light of available scientific evidence [32]. Overall, it appears that the efficacy of homeopathy is comparable to placebo, and clinical trials about open-label placebo treatments show their efficacy for the symptomatic management of some health conditions, such as pain, nausea, irritable bowel syndrome, fibromyalgia, hypertension, hyperactive bladder, insomnia, depression, some sexual dysfunctions, osteoarthritis symptoms, and the restless leg syndrome [32].

These "placebo-responsive" conditions have a different etiology, but often show a symptomatology caused by a specific functional involvement and response of the central and peripheral nervous systems. At least partially, such disease-induced responses can be modulated by the action exerted by placebo effects on the brain and, as a consequence of it, perceived symptoms can subjectively improve. Placebo effects can go beyond the sole administration of an inert substance, and also include the patient–practitioner relationship, the characteristics of the healing setting, and the rituality of care [38]. In homeopathic practice, all these factors are well represented and, therefore, the patient's response to them is elicited.

If we consider that patients with placebo-responsive conditions can benefit from placebo treatments (even open-label ones), and if we recognize that homeopathy can be considered as a very elaborated placebo treatment, then homeopathy can be reinterpreted as a safe medicinal practice with precise limits and opportunities. The use of homeopathy would become more limited if compared with how it is practiced today, but this reinterpretation would allow its safe use in specific situations without any need for a total prohibition.

## 3. Homeopathy and Psychological Therapies

Psychological therapies can be defined as a wide range of nonpharmacological counselling-based interventions, inspired by the principles of clinical psychology, which are aimed to help people in coping with distress and in improving their psychosocial functioning through the promotion of positive changes in their thinking, behaviors, and relationships [39]. According to the American Psychological Association, all psychological therapies belong to a category of the following [40]:

- Psychoanalysis and psychodynamic therapies, based on the study of the unconscious to modify problematic behaviours, feelings, and thoughts.
- Behavior therapy, focused on learning how to develop normal and correct abnormal behaviors.
- Cognitive therapy, focused on the patient's thoughts and on any dysfunctional way of thinking.
- Humanistic therapy, emphasizing the role of rationality to unlock the patient's potential and improve their functioning.

- Integrative or holistic therapy, a different combination of two or more of the above mentioned approaches depending on the practitioner's skills and on the patient's health needs.

Homeopathy has long been used for problems of a psychological nature and, recently, some theories have underscored that there are similarities between homeopathic care and some psychological therapies, including psychoanalysis [41–43]. Moreover, according to the so-called "Dodo Bird Verdict", a theory of many in the field of epistemology, different psychological and psychological-like therapies may share a broadly similar degree of clinical efficacy, regardless of their specific approach [44], and this might apply to homeopathy as well when reinterpreted as a psychotherapy. In particular, individualized homeopathy has been portrayed by some authors as a humanistic-like therapy, with influences from narrative medicine, due to its person-centered approach and detailed patient's interviews [41], and some synergies have been outlined between homeopathy and psychoanalysis, especially with regard to the characteristics of the patient–practitioner relationship [45]. Despite this, in contrast with common psychological therapies, homeopathy still retains its distinctive features as a remedy-based practice in which the subject's interview aims to understand the underlying disease roots along with specific characteristics of the patient's biotype/constitution (classified as "carbonic", "phosphoric", "fluoric", and "sulphuric" on the basis of some psycho-physical features [46]) in order to select the best remedy among the many. However, homeopathic consultations are characterized by a deep analysis of the patient's inner world of experiences and associations in such a way as to create an evocative premise for eliciting placebo effects (and the physiological response triggered by them). This can result in the creation and introduction into the patient's mind of psychologically healing inputs, possibly at an unconscious level. Historically, homeopathy is rich in symbols: Homeopathic remedies, with their Latin names, remind of very many different substances, objects, or living beings (for example, plants like *Aconitum napellus* or *Cactus grandiflorus*; minerals like *Magisterium bismuthi* or *Sulfur iodatum*; animals like *Vipera aspis* or *Blatta orientalis*), and they are all believed to possess a variable range of therapeutic "power" depending on their degree of dilution. We can possibly hypothesize that beneficial effects of homeopathy are, at least partially, due to the introduction of symbols into the patient's psychism, and their efficacy may be the same regardless of the subject's cultural awareness of them. In fact, "homeopathic symbols" may recall Jungian archetypes, that derive from the collective unconscious and are shared by all human beings in spite of their specific background [47], or they may simply offer a new identity/shape to symbols and forces already present in the patient's mind, but unable to emerge.

It is known that hypnosis-based therapeutic approaches aim to obtain beneficial effects by conveying specific "data" to the patient's mind with the help of hypnotic trance, in order to trigger a healing response and to fix "crystallized" behaviors and beliefs which may form the deepest roots of some psychological problems [48]. If we consider homeopathic care as a placebo treatment capable of stimulating some physiological effects, it is possible to understand how this potential can become the anchor to make each transferred symbol (represented by homeopathic remedies) work inside the patient's unconscious and trigger a beneficial transformation. The therapeutic rituality of repetitions in remedy administration over days and weeks during the course of the treatment can elicit placebo effects and can enhance the above-mentioned process [49]. If properly used, each symbol has an intrinsic transforming potential and can modify a static situation, thus acting as a neutralizing factor between two opposite and conflicting forces or tensions. Therefore, it is possible to imagine why many people report subjective beneficial effects from homeopathy, even if these benefits are hard to measure from a clinical/organic point of view. In fact, it can be hypothesized that homeopathy can act at an unconscious level by modifying the disease experience, characterized by an extreme inter-individual variability in its constitutive elements.

As an example, it is possible to describe the hypothetical but realistic case of a patient with anxiety who experiences recurrent nightmares and for whom it is proposed a treatment

for some weeks with *Aconitum napellus*, a homeopathic remedy obtained from a highly diluted extract of the homonym poisonous plant. Along with a placebo treatment capable of reducing anxiety symptoms, the patient may benefit from receiving a symbol or, in other words, a neutralizing factor, which can help to overcome their overwhelming feeling of powerlessness and "psychological paralysis", thus contributing to find again the lost inner serenity (as it was prior to experiencing this condition) and a good sleep.

## 4. Conclusions

If homeopathy is reinterpreted both as a placebo treatment and as a psychothera-peutic support and, on the basis of these considerations, a list of evidence-based clinical indications is formulated, then homeopathic practice may still have a safe and useful integrative role in modern medicine. The focus should be moved from homeopathy as a medicinal system which only relies on pharmacologically inert remedies to the homeopath as a practitioner capable of eliciting beneficial effects in patients affected by some health conditions, without any toxicity nor contraindications. This hypothetical interpretation may help to open a dialogue between the scientific community and radical supporters of homeopathy, provided that homeopaths are ready to reject a yet unproven vision of their practice and accept a new safer interpretation of homeopathy with some limitations, new opportunities, and still worthy of being practiced.

This work has some limitations: First of all, it is not an original/clinical study, but a brief analysis of the scientific literature with the ultimate goal of creating connections between two different fields of knowledge; then, it provides a concise literature overview and it is not meant to cover the entire topic in depth; finally, this hypothesis, although backed by available scientific evidence, should be further explored with dedicated studies aimed to quantify the actual magnitude of beneficial effects for the patient when home-opathy is reinterpreted and used as described above. In this regard, conducting clinical studies with a mixed quali-quantitative design and a population of patients affected by well-established placebo-responsive conditions/symptoms may be preferable in order to properly account for both subjective and objective health-related benefits.

**Author Contributions:** Conceptualization, D.D.; methodology, D.D. and M.A.; validation, D.D. and M.A.; investigation, D.D. and M.A.; resources, D.D. and M.A.; data curation, D.D. and M.A.; writing—original draft preparation, D.D. and M.A.; writing—review and editing, D.D. and M.A.; visualization, D.D. and M.A.; supervision, D.D. and M.A.; project administration, M.A. Both authors have read and agreed to the published version of the manuscript.

**Funding:** This research received no external funding.

**Conflicts of Interest:** The authors declare no conflict of interest.

**Entry Link on the Encyclopedia Platform:** https://encyclopedia.pub/4923.

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
