# Peer review of "Homeopathy and Psychological Therapies"

_encyclopedia, doi:10.3390/encyclopedia1010008_

Round 1
Reviewer 1 Report
The Authors of the presented study draw attention to a significant number of patients and doctors using homepathic methods of treatment. On the basis of the current practices related to homeopathy, they try to explain from a scientific point of view, the positive effects of therapy reported by the supporters of homeopathy. They take into account the effect of placebo, well described in the literature, but also hypothesize that homeopathic treatment regimens are a kind of rituals that can help discover or stimulate the possibilities of the patient's psyche, especially in those disease entities where our knowledge confirms its important role. The article is interesting and can help to organize the discussion related to homeopathic treatments and their efficacy in patients.
Author Response
We would like to thank the reviewer for the appreciation of our work.
Reviewer 2 Report
The reviewed report discusses homeopathy and psychological therapies.
Although it is a brief summary, the following key points should be considered:
Definition and key words
The abstract should emphasize the main points regarding the state of the issue. In this case what was and what is meant by homeopathy. Also, although reference is made to the purpose, some conclusions should also be included.
The key words are excessive and some of them are unrelated to the topic. Sustainability? Scientific evidence? Safety? Symbol?
Introduction
A historical approach to the term homeopathy and a clear definition are missing. The same applies to the term psychological therapy. These are essential aspects that would make the review more solid.
Scientific evidence and placebo effect
The authors point out only 3 references in the scientific evidence section. A deep reform of the section is necessary, including more outstanding references in the topic.
Homeopathy and psychological therapy
In this section it is necessary to go into the issue in greater depth. There are aspects that remain unresolved. Why is homeopathy associated with psychological therapies? Is homeopathy not used in terminal illnesses? What is your opinion of homeopathy in the scientific world? Are there detractors and defenders of this type of therapy?
Finally, in the reference section, it should be reviewed as there are some errors.
Author Response
Overall, we have tried to address the reviewer’s concerns point by point, but we had to keep the manuscript as synthetic as possible to comply with editorial standards and with the journal aim.
Definition and key words
The abstract should emphasize the main points regarding the state of the issue. In this case what was and what is meant by homeopathy. Also, although reference is made to the purpose, some conclusions should also be included.
REPLY: We had to shorten the abstract and turn it into a “Definition” section due to an explicit request of the editor in order to meet the journal standards.
The key words are excessive and some of them are unrelated to the topic. Sustainability? Scientific evidence? Safety? Symbol?
REPLY: Following the reviewer’s suggestion, some loosely related keywords have been erased.
Introduction
A historical approach to the term homeopathy and a clear definition are missing. The same applies to the term psychological therapy. These are essential aspects that would make the review more solid.
REPLY: Following the reviewer’s suggestion, we have added a paragraph with a brief history and definition of homeopathy in the “Introduction” section. Additionally, we have reported the definition and categorization of psychological therapies in the third section of the manuscript.
Scientific evidence and placebo effect
The authors point out only 3 references in the scientific evidence section. A deep reform of the section is necessary, including more outstanding references in the topic.
REPLY: The issue of homeopathy efficacy is quite complex, and, in order to effectively summarize the topic, we have only cited extensive high-quality umbrella reviews, namely systematic reviews of systematic reviews, thus providing the reader with specific readings to cover this topic in sufficient depth. We have now added more references, thus outlining the position of some national healthcare institutions on the basis of existing evidence.
Homeopathy and psychological therapy
In this section it is necessary to go into the issue in greater depth. There are aspects that remain unresolved.
Why is homeopathy associated with psychological therapies?
REPLY: We have explained why a similarity exists in this section of the manuscript.
Is homeopathy not used in terminal illnesses?
REPLY: It is prescribed even for incurable illnesses (see references 14-16), but its use has to be avoided as an alternative therapy. Our paper aims to propose a more rational and safer use of homeopathy as a complementary/integrative aid with beneficial effects on the patient’s mind and wellbeing.
What is your opinion of homeopathy in the scientific world?
REPLY: Our opinion is well explained by this extensive literature review of ours, that we have mentioned and summarized in the manuscript: https://doi.org/10.1111/hsc.12681
Are there detractors and defenders of this type of therapy?
REPLY: Yes, there are, as mentioned in the introduction section of the paper (4th paragraph).
Finally, in the reference section, it should be reviewed as there are some errors.
REPLY: The reference section has been checked and formatted again with the help of a dedicated software (Paperpile).
Round 2
Reviewer 2 Report
The paper has had an extensive modification and is ok for publication. Good work